# Vial Sharing of High-Cost Drugs to Decrease Leftovers and Costs: A Retrospective Observational Study on Patisiran Administration in Bologna, Italy

**DOI:** 10.3390/healthcare11071013

**Published:** 2023-04-02

**Authors:** Margherita Cozzio, Alessandro Melis, Giusy La Fauci, Pietro Guaraldi, Rosaria Caputo, Flavia Lioi, Giulia Sangiorgi Cellini, Giuseppina Santilli, Donatella Scarlattei, Pasquale Siravo, Paola Zuccheri, Andrea Ziglio, Marco Montalti

**Affiliations:** 1Unit of Hygiene and Medical Statistics, Department of Biomedical and Neuromotor Sciences, University of Bologna, 40126 Bologna, Italy; 2Unit of Hospital Management, IRCCS Institute of Neurological Sciences of Bologna, 40139 Bologna, Italy; 3Unit of Clinical Neurology NEUROMET, IRCCS Institute of Neurological Sciences of Bologna, 40139 Bologna, Italy; 4Pharmacy Unit for Cytotoxic Drug Preparations, Bellaria Hospital, Intercompany Pharmaceutical Department, Bologna Local Health Authority, 40139 Bologna, Italy; 5IRCCS Institute of Neurological Sciences of Bologna, 40139 Bologna, Italy; 6Unit of Hospital Management, Arco and Tione Hospitals, APSS Trento, 38100 Trento, Italy

**Keywords:** drug compounding, patisiran, hATTR amyloidosis, drug day, sustainability, pharmaceutical spending, medication waste, orphan drugs, vial sharing

## Abstract

Waste of high-cost medicines, such as orphan drugs, is a major problem in healthcare, which leads to excessive costs for treatments. The main objective of this study was to evaluate the impact of a vial-sharing strategy for patisiran, an orphan drug used for the treatment of hereditary transthyretin-mediated amyloidosis, in terms of a reduction in the discarded drug amount and cost savings. The retrospective observational study was conducted in a tertiary referral center (Emilia-Romagna, Italy), between February 2021 and November 2022. Data on drug waste were calculated as “(mg used–mg prescribed)/mg prescribed” for each session. We found a statistically significant (−9.14%, *p* < 0.001, 95% CI 5.87–12.41) absolute difference in mean discarded drug rates per session based on the study phase (before and after vial-sharing introduction) at the two-sample t-test. The absolute difference corresponded to a percentage decrease in the average reduction in the discarded drug rate with vial sharing of 82.96% per session. On an annual scale, the estimated cost savings was EUR 26,203.80/year for a patient with a standard body weight of 70 kg. In conclusion, we demonstrated that a patisiran vial-sharing program undoubtedly offsets some of the high costs associated with this treatment. We suggest that this easy-to-introduce and cost-effective approach can be applied to the administration of other high-cost drugs.

## 1. Introduction

The Italian National Health Service (NHS) was established in 1978 and is based on the equity principle and inspired by the 1948 British system [1]. The NHS is highly decentralized, and each Italian region, overseen at the State level, is responsible for organizing and delivering health services to the population. The central government allocates funds for regional health systems, and, in order to mitigate the risk of inequalities, establishes all the services and benefits that the NHS is required to provide to all citizens [2,3].

According to the report of the Organisation for Economic Co-operation and Development (OECD), in 2019, Italy spent 8.7% of its gross domestic product (GDP) on healthcare, equal to EUR 154.8 billion, financed mainly from the State budget (63% in 2018), i.e., essentially through value-added taxes (VATs) and excise duties on fuel and through the National Health Fund [4].

Pharmaceutical expenditure is a major component of public spending on healthcare: it accounts for almost 21% of the economic resources that are annually committed to healthcare, in 2019, amounting to 1.7% of the national GDP, i.e., EUR 30.8 billion [2,5].

The national authority for pharmaceutical regulation is the Italian Medicines Agency (AIFA), which authorizes clinical trials and approves drugs that can be produced and marketed in Italy. AIFA assesses the cost-effectiveness and sustainability of drugs for the NHS and divides them into different reimbursement categories [2].

Some current and emerging therapies are associated with significant costs and can be a barrier to patient care [6].

An example of high-cost therapies are drugs for the treatment of life-threatening or chronically debilitating rare diseases, with a European Union prevalence being no more than 5 in 10,000 inhabitants (orphan drugs) [7], that do not have a sufficient market to repay the cost of their development [8,9,10].

In 2019, the total expenditure on the 71 orphan drugs authorized in Italy was approximately EUR 1.5 billion, representing 6.6% of the pharmaceutical expenditure charged to the Italian NHS [11].

To contain pharmaceutical expenditure, monitoring and governance tools, such as spending caps, have been progressively introduced in the country [12,13,14].

At the hospital level, pharmaceutical expenditure can be controlled by, for example, reducing drug wastage [15,16].

Medicine waste refers to any pharmaceutical product that remains unused or is not fully consumed throughout the pharmaceutical supply and use chain [17].

Incorrect inventory management, lengthy procurement cycles, poor storage, improper monitoring of drug expiration times, distribution problems, and irrational usage of drugs result in wastage of pharmaceuticals [18].

All the various stakeholders involved in the pharmaceutical chain (manufacturers, distributors, prescribers, pharmacists, patients, and health authorities) can prevent the waste of potentially viable medications [19]. As for pharmacists, their role is of particular importance at the drug-compounding stage. Indeed, drugs could be discarded if vials are packed in a single size that is different from the prescribed dose based on the patient’s weight [20]. In addition, once the vials are opened, they must be administered or discarded [21]. Leftover medication must be paid for, even if discarded.

Patisiran is an example of a high-cost single-dose orphan drug with a patient weight-dependent dosage that must be administered for life. It is indicated for the treatment of hereditary transthyretin-mediated amyloidosis (hATTR amyloidosis) in adult patients with stage 1 or stage 2 polyneuropathy [22]. Patisiran was the first therapy based on FDA-approved short-interfering RNA (Si-RNA) technology [22,23] and acts by reducing the production of mutant and wild-type transthyretin, thus improving the multiple clinical manifestations of hATTR amyloidosis [24]. Patisiran is included in the list of Class H drugs, completely reimbursed by the NHS and only distributable by (or used within) hospitals [2]

There are several mechanisms and tools that could be used to reduce discarded drug amounts and thus pharmaceutical expenditure [17,19], leading to quality improvement through the redirection of resources to add value to patient care [15].

One approach is the vial-sharing procedure, described as the process of making each dose separately and keeping the leftover of the last-used vial to be reused in the production of the subsequent patient, dose-treated during one session in the same center on the same day [25].

This study aims to present an economically viable model for patisiran administration and quantify the percentage reduction of the discarded drug amount per administration session and cost savings by introducing vial sharing.

## 2. Materials and Methods

### 2.1. Study Design, Sample, and Procedure

This is an observational retrospective study conducted at IRCCS Institute of Neurological Sciences of Bologna, a tertiary referral center (Emilia-Romagna, Italy), between February 2021 and November 2022. No randomization or special selection was carried out. All participants provided informed consent to being included in the study. As the study had an anonymous, observational design and was not a clinical trial, a preliminary evaluation by an Ethical Committee/Institutional Review Board was not required, according to Italian law (Gazzetta Ufficiale no. 76, dated 31 March 2008).

All subjects living with hATTR amyloidosis treated with patisiran and followed by neurologists from two different departments within the Institute of Neurological Sciences of Bologna were included in the study. Patisiran was intravenously infused to each patient at a dosage of 0.3 mg/kg of body weight (KBW) every three weeks. The drug was supplied in single-use 10 mg/5 mL vials, and it was sold at an ex-factory price of EUR 8529.41/vial. General points considered for contemplating vial sharing of patisiran are summarized in Table 1.

Data on the total weight (in kg) of patients treated in a session and the amount (in mg) of drug used in each session were collected during two different phases corresponding to before and after the introduction of vial-sharing practice:Phase 1: from February 2021 to March 2022. Staff from two different outpatient clinics belonging to the Institute were independently responsible for recruiting patients. The administration of the drug to the enrolled patients was managed on separate days.Phase 2: from April 2022 to November 2022. Patient management of the Institute’s two outpatient clinics was entrusted to the Pharmaceutical Unit. The staff in charge divided patients into two groups according to their body weight and arranged drug administration sessions for each group. The sum of the total body weight of the two neo-groups was a value very close to a multiple of 33 (each vial covers 33.3 KBW). Within each group, the appropriately stored surplus of a vial was used for the compounding for the next patient.

In both phases, the Centralized Pharmaceutical Unit was responsible for compounding the drug and reporting any leftovers. Compounding centralization and strict adherence to the preparation norms resulted in a guarantee in terms of safety not only for those who receive the treatment but also for those in charge of compounding, ensuring the quality and sterility of the final product.

In the vial-sharing procedure, the compounding and the administration of patisiran treatments on a single day for each group were unified according to the schedule once every three weeks. We collected data on the number of patisiran vials used and the amount (in mg) of patisiran administered to each patient in each treatment session.

### 2.2. Statistical Analysis

Numerical variables were summarized as mean ± standard deviation; categorical variables were summarized as frequencies and percentages. Drug amounts were always expressed in mg, subjects’ body weight in kg, and drug prices/cost savings in euros.

Discarded drug rate was calculated for each session as follows: (mg used–mg prescribed)/mg prescribed. A two-sample *t*-test was run to determine if there were differences in discarded drug rates per session based on the study phase, consisting of a Phase 1 (pre-vial sharing) and Phase 2 (post vial sharing). All analyses were carried out using Stata Statistical Software 17 [26]. Microsoft Excel 2016 was used for data collection.

## 3. Results

### 3.1. Sample

The sample consisted of 18 subjects: 13 patients participated in Phase 1 and 14 in Phase 2; 4 Phase 1 subjects left the study early. In the two different phases, enrolled subjects were 61.1% and 50.0% males, respectively. The mean age was 62.6 ± 16.4 years in Phase 1 and 61.4 ± 17.0 years in Phase 2. Mean body weight was 66.1 ± 15.8 kg in Phase 1 and 67.9 ± 17.6 kg in Phase 2. All patients’ details are shown in Table 2.

### 3.2. Vial Sharing and Discarded Drug Savings

All information on the number of patisiran administration sessions, the number and weight of patients involved per session, and the amount of drug used, prescribed, and discarded is shown in Table 3.

From February 2021 to November 2022, 99 patisiran administration sessions were organized: 78 before the introduction of vial sharing and 21 afterward. During Phase 1, an average of 1.76 patients ± 0.63 per session participated with a total average weight of 100.61 ± 37.85 kg, while in Phase 2, patients per session were 5.86 ± 1.01, for a total of 364.24 ± 86.06 kg on average per session.

The average amount of discarded drug during Phase 1 sessions was 3.15 ± 1.95 mg and 2.00 ± 2.30 mg during Phase 2 sessions against an average prescribed amount of 30.18 ± 11.35 mg and 108.95 ± 25.80 mg, respectively.

In contrast, the discarded drug rate, calculated as: “(mg used–mg prescribed)/mg prescribed”, was 11.01 ± 7.45% before the introduction of vial sharing and 1.88 ± 2.15% after the introduction of the procedure (see Table 3).

The results of the two-sample *t*-test show that the Phase 2 sessions had statistically significant lower discarded drug rates (1.88 ± 2.15%, 95% CI 9.33–12.69) compared to the Phase 1 sessions (11.01 ± 7.45%, 95% CI 0.90–2.85), *t*(95) = 5.543, *p* < 0.001 (See Table 4) with an absolute reduction of 9.14% with a 95% confidence interval of 5.87–12.41.

This absolute reduction corresponded to an average percentage reduction in the discarded drug rate between the two phases of 82.96% per session.

### 3.3. Cost Savings

Considering the price per vial of EUR 8529.41 (EUR 852.94 per mg) and the amount of discarded drugs per session (see Table 3), a total amount of patisiran equivalent to EUR 209,610.25 in Phase 1 and EUR 35,908.82 in Phase 2 was discarded, with a monthly average of EUR 14,972.16 and EUR 4488.62, respectively. In Phase 1, an average of EUR 26.71/KBW/session was spent on discarded drugs, while in Phase 2, discarded drugs accounted for EUR 4.69/KBW/session.

Vial sharing resulted in a saving of EUR 22.02/KBW/session. On an annual scale, the estimated savings are EUR 26,203.80/year for a standard 70 KBW patient.

## 4. Discussion

Our study showed that the introduction of vial sharing in high-cost orphan drug administration with a patient weight-dependent dose (patisiran) led to a statistically significant average reduction in the discarded drug rate of 82.96% per session.

A cost saving of EUR 26,203.80/year for a standard 70 KBW patient was estimated after the introduction of vial sharing, pointing out the real economic impact of this method in our study. The findings suggest that any treatment with similar characteristics would be optimal for applying vial sharing as a dose and cost rationalization strategy.

Designing strategies to reduce healthcare costs and drug waste is one of the main goals of all healthcare professionals. In a State with a predominantly public healthcare system, such as Italy [27], simple resource optimization strategies, such as vial sharing, are of great benefit to the entire community.

In this context, our study has shown how the use of simple strategies of resource optimization, such as vial sharing, is effective in reducing the rate of discarded drugs and consequently treatment-related costs, especially in the case of high-cost drugs, which can lead to economic and environmental benefits and an increase in the number of people treated.

In a similar study, Smith et al. had previously analyzed different methods of chemotherapy vial sharing (on a daily, per calendar week, and a rolling seven-day basis) comparing them with no sharing at all. Again, vial sharing was shown to always lead to a reduction in discarded drugs, a decrease that was all the greater the more sharing was involved [18]. In another multicenter study conducted in Italy on the use of ipilimumab vial sharing, the price for treating a model patient was significantly lower, generating significant economic savings to be reallocated to other expenses [28]. Similarly, another retrospective study on the bevacizumab cost/benefit found a 97.88% reduction in the total annual cost after the introduction of vial sharing [29].

In a recent study by Liu et al. evaluating the potential for hospitals in China to employ a real-time vial-sharing strategy using a robot, this new technology was found to be of great help in saving up to ~59.08% of the total amount wasted of 24 different drugs [30].

Another point that arises from our results concerned the importance of proper vial size selection by the pharmaceutical industries. If vials of different sizes were produced so that the dose could be adjusted to different patient weights or if the amount packed in each single-dose vial were as close as possible to the dose needed to treat patients, discarded drugs would be greatly reduced [31].

Vial sharing is not the only strategy available to reduce drug waste and associated costs. Dose rounding, for example, involves rounding drug doses to the nearest vial size when the difference is less than an established percentage (within 5–10%) [32] and could be particularly important for drugs supplied in single-use vials in a preservative-free formulation [32]. However, when dose rounding is carried out in the lower range, it leads to the administration of an amount of drug that may be less than the patient needs. Sharing vials could help in mitigating this issue.

Moreover, pharmaceutical companies often fill vials of injectable drugs with slightly more than the nominal capacity (overfill) to ensure proper withdrawal and dosing to the patient [33]. Therefore, to further reduce healthcare spending on high-cost drugs, the potential excess volume of the injectable drug compared to the company’s stated dose could also be combined with the vial-sharing method [34]. However, it is easy to see how relying on overfilling may not be a sound way to structure cost savings, as it is not necessarily provided in all drug vials.

We recognize that our work has limitations. First, the study was conducted in a tertiary referral center for the disease; therefore, the benefits of vial sharing might be lower in hub centers with fewer users and smaller health facilities. In addition, we recognize that Phase 2 monitoring is of limited duration and that savings could be affected by changes in patient doses over time and other market forces, such as changes in the price of patisiran (although we are not aware of any plans to change the price of patisiran soon). Finally, costs arising from the activity of the Pharmaceutical Unit staff were not analyzed in our study in either phase.

## 5. Conclusions

This study demonstrated a statistically significant reduction in terms of the discarded drug amount and cost per session, introducing vial sharing as a cost-effective model for a patient weight-dependent dose for orphan drug administration. Vial sharing is confirmed as an easy-to-introduce strategy with proven effectiveness in reducing spending without altering the quality of care. Given that the expenditure associated with some current and emerging therapies (including orphan drugs) could be a barrier to patient care, we suggest that this approach could be applied to other high-cost drugs.

## Figures and Tables

**Table 1 healthcare-11-01013-t001:** Suitable characteristics of patisiran for vial sharing.

Drug Characteristic	Patisiran
Expensive drug	Cost of one vial is EUR 8529.41
Packaged in a single size	Supplied in 10 mg/5 mL vials only
Supplied in single-use vials	Should be used immediately, and any drug residues must be disposed of. If not used immediately, store in the infusion bag at room temperature (up to 30 °C) for up to 16 h (including infusion time).
Dosage based on patient’s weight	Administered at a dosage of 0.3 mg per KBW
Well-defined patient population	Indicated for a lifelong treatment of Polyneuropathy in adult patients with hATTR amyloidosis

*Notes*: mg, milligrams; mL, milliliter; KBW, Kilograms of Body Weight; °C, degree Celsius; hATTR amyloidosis, hereditary transthyretin-mediated amyloidosis (hATTR).

**Table 2 healthcare-11-01013-t002:** Sociodemographic characteristics of all patients (N = 18) in Phase 1 and Phase 2 of the study.

	Phase 1 (N = 13)			Phase 2 (N = 14)		
Patient	Gender	Age (years)	Weight (kg)	Gender	Age (years)	Weight (kg)
1	M	76	56	M	77	56
2	M	51	64	M	52	64
3	F	32	53	F	32	53
4	F	47	53.5	F	47	53.5
5 *	M	62	63			
6 *	M	81	76			
7 *	M	62	76			
8	M	70	63	M	71	63
9	F	40	49	F	41	49
10 *	M	57	100			
11	F	81	48	F	82	48
12	M	77	90	M	78	90
13	F	78	68	F	79	68
14 **				F	65	55
15 **				M	50	98
16 **				F	41	84
17 **				M	77	71
18 **				M	68	98
% or Mean	61.5% (M)	62.6	66.1	50.0% (M)	61.4	67.9
SD		16.4	15.8		17.0	17.8

*Notes*: N, number of participants; * left the study after Phase 1; ** enrolled in Phase 2; M, Male; F, Female; SD, Standard Deviation.

**Table 3 healthcare-11-01013-t003:** Sessions’ characteristics in Phase 1 (a) and Phase 2 (b). Number of treated patients, total cumulative KBW of treated patients, amount of used drug (in mg), amount of drug prescribed by physicians (in mg), amount of discarded drug (in mg), and the percentage of discarded drug compared to prescribed drug are reported for each session.

**a.**							
	**Session**	**Treated Patients (N)**	**Total KBW (kg)**	**Used Drug (mg)**	**Prescribed Drug (mg)**	**Discarded Drug (mg)**	**Discarded Drug (%)**
Phase 1	1	1	56	20	16.80	3.20	19.05
	2	1	64	20	19.20	0.80	4.17
	3	2	96.5	30	28.95	1.05	3.63
	4	1	56	20	16.80	3.20	19.05
	5	1	64	20	19.20	0.80	4.17
	6	2	96.5	30	28.95	1.05	3.63
	7	1	56	20	16.80	3.20	19.05
	8	1	64	20	19.20	0.80	4.17
	9	1	63	20	18.90	1.10	5.82
	10	1	76	30	22.80	7.20	31.58
	11	2	96.5	30	28.95	1.05	3.63
	12	1	56	20	16.80	3.20	19.05
	13	1	64	20	19.20	0.80	4.17
	14	1	63	20	18.90	1.10	5.82
	15	2	96.5	30	28.95	1.05	3.63
	16	1	56	20	16.80	3.20	19.05
	17	1	64	20	19.20	0.80	4.17
	18	2	139	50	41.70	8.30	19.90
	19	2	96.5	30	28.95	1.05	3.63
	20	1	56	20	16.80	3.20	19.05
	21	1	64	20	19.20	0.80	4.17
	22	2	92.5	30	27.75	2.25	8.11
	23	1	53	20	15.90	4.10	25.79
	24	2	119	40	35.70	4.30	12.04
	25	1	64	20	19.20	0.80	4.17
	26	2	93.5	30	28.05	1.95	6.95
	27	1	53	20	15.90	4.10	25.79
	28	2	119	40	35.70	4.30	12.04
	29	1	64	20	19.20	0.80	4.17
	30	1	100	30	30.00	0.00	0.00
	31	2	93.5	30	28.05	1.95	6.95
	32	1	53	20	15.90	4.10	25.79
	33	2	119	40	35.70	4.30	12.04
	34	1	64	20	19.20	0.80	4.17
	35	2	119	40	35.70	4.30	12.04
	36	2	93.5	30	28.05	1.95	6.95
	37	2	115	40	34.50	5.50	15.94
	38	2	138	50	41.40	8.60	20.77
	39	2	119	40	35.70	4.30	12.04
	40	2	93.5	30	28.05	1.95	6.95
	41	2	115	40	34.50	5.50	15.94
	42	2	125	40	37.50	2.50	6.67
	43	3	179	60	53.70	6.30	11.73
	44	2	93.5	30	28.05	1.95	6.95
	45	2	115	40	34.50	5.50	15.94
	46	2	123	40	36.90	3.10	8.40
	47	3	184.5	60	55.35	4.65	8.40
	48	2	93.5	30	28.05	1.95	6.95
	49	2	115	40	34.50	5.50	15.94
	50	2	124	40	37.20	2.80	7.53
	51	3	184	60	55.20	4.80	8.70
	52	2	93.5	30	28.05	1.95	6.95
	53	1	64	20	19.20	0.80	4.17
	54	2	125	40	37.50	2.50	6.67
	55	1	51	20	15.30	4.70	30.72
	56	3	184	60	55.20	4.80	8.70
	57	2	93.5	30	28.05	1.95	6.95
	58	1	64	20	19.20	0.80	4.17
	59	2	125	40	37.50	2.50	6.67
	60	1	51	20	15.30	4.70	30.72
	61	3	185	60	55.50	4.50	8.11
	62	2	91.5	30	27.45	2.55	9.29
	63	2	125	40	37.50	2.50	6.67
	64	3	185	60	55.50	4.50	8.11
	65	2	115	40	34.50	5.50	15.94
	66	2	91.5	30	27.45	2.55	9.29
	67	2	125	40	37.50	2.50	6.67
	68	2	119	40	35.70	4.30	12.04
	69	3	176	60	52.80	7.20	13.64
	70	2	91.5	30	27.45	2.55	9.29
	71	2	125	40	37.50	2.50	6.67
	72	2	119	40	35.70	4.30	12.04
	73	2	125	40	37.50	2.50	6.67
	74	1	51	20	15.30	4.70	30.72
	75	2	91.5	30	27.45	2.55	9.29
	76	2	125	40	37.50	2.50	6.67
	77	2	119	40	35.70	4.30	12.04
	78	3	176	60	52.80	7.20	13.64
Mean		1.76	100.61	33.33	30.18	3.15	11.01
SD		0.63	37.85	12.45	11.35	1.95	7.45
**b.**							
	**Session**	**Treated Patients (N)**	**Total KBW (kg)**	**Used Drug (mg)**	**Prescribed Drug (mg)**	**Discarded Drug (mg)**	**Discarded (%)**
Phase 2	79	5	295	90	88.50	1.50	1.69
	80	4	216.5	70	64.95	5.05	7.78
	81	5	295	90	88.50	1.50	1.69
	82	5	295	90	88.50	1.50	1.69
	83	6	369.5	110	110.00	0.00	0.00
	84	6	369.5	110	110.00	0.00	0.00
	85	5	295	90	88.50	1.50	1.69
	86	5	295	90	88.50	1.50	1.69
	87	5	295	90	88.50	1.50	1.69
	88	6	369.5	110	110.00	0.00	0.00
	89	5	288.5	90	86.55	3.45	3.99
	90	6	369.5	110	110.00	0.00	0.00
	91	6	379	120	113.70	6.30	5.54
	92	7	449	140	134.70	5.30	3.93
	93	6	369.5	110	110.00	0.00	0.00
	94	8	547	170	164.10	5.90	3.60
	95	6	369.5	110	110.00	0.00	0.00
	96	6	369.5	110	110.00	0.00	0.00
	97	8	547	170	164.10	5.90	3.60
	98	7	496	150	148.80	1.20	0.81
	99	6	369.5	110	110.00	0.00	0.00
Mean		5.86	364.24	110.95	108.95	2.00	1.88
SD		1.01	86.06	26.63	25.80	2.30	2.15

*Notes*: N, number of treated patients; mg, milligrams; KBW, Kilograms of Body Weight; kg, Kilograms; SD, Standard Deviation.

**Table 4 healthcare-11-01013-t004:** Two-sample *t*-test for differences in discarded drug rates per session in the two study phases.

Phase	Sessions (N)	Mean Discarded Drug (%)	SD	95% CI
1	78	11.01	7.45	9.33–12.69
2	21	1.88	2.15	0.90–2.85
diff		9.14		5.87–12.41
t = 5.54; Pr(|T| > |t|) = 0.0000				

*Notes*: N, number of sessions; SD, Standard Deviation; CI, Confidence Interval; diff, difference; t, *t*-test.

## Data Availability

The data presented in this study are available on request from the corresponding author.

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
