# Peer review of "Vial Sharing of High-Cost Drugs to Decrease Leftovers and Costs: A Retrospective Observational Study on Patisiran Administration in Bologna, Italy"

_healthcare, 2023, doi:10.3390/healthcare11071013_

Round 1

Reviewer 1 Report

The total healthcare expenditure in Italy should be mentioned in the opening paragraph, I believe you will find this information in the Eurostat database.

None of the tables have a legend explaining symbols and abbreviations.

Reviewer 2 Report

The authors provide a vial-sharing model for reducing discarded drug amounts for an orphan disease in the manuscript. 

The approach is reasonable, but the quality of the manuscript is concerning. 

-Please remove statistical details like the t-test from the abstract, and decipher abbreviations like KBW. Also, the names of the centers and timelines are not essential for the readers. The last sentence also should be removed from the abstract, so I recommend rewriting it. 

-Please also add a flow diagram of the study design

-Table 3 is hard to understand; please remake it or add it to the supplementary. Also, please improve table 3, so readers can make some conclusions, not just make an observation.

Reviewer 3 Report

This study shows that the introduction of vial sharing in high-cost orphan drug use with patisiran leads to a significant reduction in discarded drug rate. Statistical analysis is quite fine, and finds a cost saving of 23,204 euros per year for a standard patient. The work has some limitations with phase 2 monitoring being of limited duration and the tertiary referral center being not a hub center (fewer users). Some mistakes in English should be corrected. For instance, "expensivedrug" and "wheight" (sic).
